# Vortex fluidics-mediated DNA rescue from formalin-fixed museum specimens

**Christian A. Totoiu**[1¤], **Jessica M. Phillips**[2], **Aspen T. Reese**[3], **Sudipta Majumdar**[1], **Peter R. Girguis**[3], **Colin L. Raston**[2], **Gregory A. Weiss**[1,4,5]*

**1** Department of Chemistry, University of California, Irvine, California, United States of America, **2** Flinders Institute for Nanoscale Science and Technology, College of Science and Engineering, Flinders University, Adelaide, South Australia, Australia, **3** Department of Organismic and Evolutionary Biology, Harvard University, Cambridge, Massachusetts, United States of America, **4** Department of Molecular Biology and Biochemistry, University of California, Irvine, California, United States of America, **5** Department of Pharmaceutical Sciences, University of California, Irvine, California, United States of America

¤ Current address: Department of Chemical Engineering and Biotechnology, University of Cambridge, Cambridge, England, United Kingdom
* gweiss@uci.edu

**Data Availability Statement:** All relevant data are within the manuscript and its Supporting Information files.

## Abstract

DNA from formalin-preserved tissue could unlock a vast repository of genetic information stored in museums worldwide. However, formaldehyde crosslinks proteins and DNA, and prevents ready amplification and DNA sequencing. Formaldehyde acylation also fragments the DNA. Treatment with proteinase K proteolyzes crosslinked proteins to rescue the DNA, though the process is quite slow. To reduce processing time and improve rescue efficiency, we applied the mechanical energy of a vortex fluidic device (VFD) to drive the catalytic activity of proteinase K and recover DNA from American lobster tissue (*Homarus americanus*) fixed in 3.7% formalin for >1-year. A scan of VFD rotational speeds identified the optimal rotational speed for recovery of PCR-amplifiable DNA and while 500+ base pairs were sequenced, shorter read lengths were more consistently obtained. This VFD-based method also effectively recovered DNA from formalin-preserved samples. The results provide a roadmap for exploring DNA from millions of historical and even extinct species.

## Introduction

Archived biological samples offer an important source of genetic information for diverse fields including evolutionary biology, ecology, phylogenetics, biodiversity, and epidemiology [1–2]. Samples, from hydrated tissues to whole organisms, have historically been preserved in aqueous formaldehyde (3.7 to 4% solution of formaldehyde in water, termed formalin). In many cases, these specimens are the only remaining samples that could provide genetic information about the organisms, including their microbiomes, environments, diets, and other attributes– all from the moment of sample preservation [3–5]. This preservative, however, hinders DNA amplification and sequencing with the sample [6]. Thus, new methods to recover DNA from formalin-fixed specimens could advance our ability to access the genetic information in these samples, and advance our understanding of how organisms and ecosystems have responded to

**Funding:** This work was supported by the National Human Genome Research Institute (NHGRI, https://www.genome.gov) of the National Institutes of Health [1R01HG009188-01 to G.A.W.]; the Star Family Foundation Challenge for Promising Scientific Research [to A.T.R. and P.R.G.]; and the University of California at Irvine's Undergraduate Research Opportunities Program and Summer Undergraduate Research Program [to C.A.T.]. Funding for open access charge: NHGRI. The funders had no role in study design, data collection and analysis, decision to publish, or preparation of the manuscript.

natural and anthropogenic changes over time. For example, formalin-fixed specimens in natural history museums could be used to elucidate the impact of environmental changes on the DNA of biological populations [1–2, 7]. DNA sequencing of such samples could address longitudinal, biological questions that may be impractical to address without the genetic information for the preserved specimens [2, 8–9].

For >150 years, formalin fixation has been used to effectively preserve hydrated specimens [7]. A vast repository of formalin-fixed samples exists, including at least 400 million samples at 13 large institutions [1]. Marine organisms are particularly well-preserved in this aqueous preservative, as it retains morphological features well, enabling more detailed taxonomic studies. Aqueous formaldehyde is also advantageous in that it stops parasitic microbial growth [10]. However, preserving samples in formalin fixation damages DNA [11–12]. Covalent modification of DNA bases by the electrophilic formaldehyde drives base deglycosylation, and the resultant abasic sites in DNA can cause strand breakage [13]. Additionally, long duration storage often incurs DNA fragmentation, independent of formalin [14]. Fragments from both mechanisms increase the amount of DNA template required for PCR amplification of longer targets, and can also inhibit PCR [15–16].

Intrastrand and protein-DNA crosslinks introduced by formaldehyde can also block PCR and DNA sequencing [17–18]. Protein-DNA crosslinks result from nucleophilic attack on formaldehyde by proteins' primary amines to yield imines and iminium ions. These groups can then react with the less nucleophilic primary amines of DNA bases, particularly from guanine, resulting in a protein-DNA crosslink (Fig 1) [19–20]. Due to the high density of amines found on the surface of proteins and DNA, each DNA-protein complex can become crosslinked multiple times. Additionally, formalin-fixed cells cannot repair the slow process of cytosine deamination to uracil [11, 21]. During PCR amplification, adenine can be incorporated as the incorrect complement to degraded cytosine, resulting in point mutations [11]. In summary, DNA damage caused by preservation results in short templates for PCR and low-quality, error-prone DNA sequences.

Despite the immense challenges that are associated with formalin fixation, much effort has been dedicated to developing techniques for sequencing these irreplaceable samples (DNA recovery methods from formalin-fixed tissue are summarized in Table 1). Most current methods for recovering DNA from formalin-fixed organisms use proteinase K, a thermostable

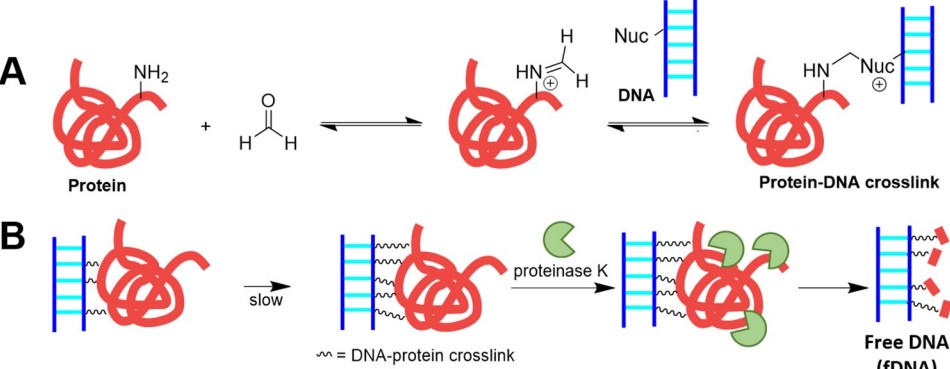

**Fig 1. Schematic of formalin-induced crosslink formation and the removal of crosslinked proteins by treatment with proteinase K. (A)** A protein amine can nucleophilically attack the formaldehyde carbonyl to yield an iminium ion, which can then react with another primary amine from DNA, RNA, or proteins to form a crosslink. This crosslink reaction is in reversible dynamic equilibrium [10, 21]. **(B)** Treatment with a protease, proteinase K, allows free DNA (fDNA) recovery. Here, Nuc designates an amine nucleophile from the DNA.

**Table 1. Current methods for formaldehyde crosslink removal & DNA recovery from formalin-fixed specimens.**

| Method | Temperature (°C) | Time (h) |
|---|---|---|
| proteinase K treatment[26] | 56 | ~17 |
| proteinase K treatment in Tris-NaCl-EDTA-SDS buffer[27] | 55 | 68 |
| hot alkali buffer treatment[7][25] | 100 to 120 | 0.4 to 0.7 |
| Cetyltrimethylammonium bromide (CTAB) & proteinase K[28] | 65 & 56 | 0.5 & 1–72 |
| QIAamp DNA Mini Kit[28] | 56 | not reported |
| QIAamp DNA FFPE Kit[28] | 56 & 90 | not reported |

serine protease with broad substrate specificity [22], to digest crosslinked proteins and eliminate most crosslink-associated blockages [21–23]. However, even at the enzyme's optimum temperature (49 ± 2 °C), the free DNA (fDNA) recovery rate from this method is low at approximately 4.4% per hour; additionally, the enzyme's half-life is limiting at approximately 11.3 h [21–22]. At room temperature (≈22 °C), the proteolytic reaction rate yields only 1.1% fDNA per hour [24]. DNA can also be recovered from formalin-fixed tissue with a 0.1 M NaOH (pH 12) buffer treatment at 120 °C for 25 min [7, 25]. However, these harsh conditions can further damage the DNA through Brønsted base-caused strand cleavage; therefore, this approach is most valuable in cases where there is an excess of tissue to be digested, and is highly inappropriate for most delicate, longer-preserved samples. Notably, the current reactions to liberate DNA are harsh, low-throughput, low yielding, and time consuming. Mild methods to increase DNA recovery and purification for PCR amplification and subsequent DNA sequencing could revolutionize the study of a wide range of museum specimens.

We posit that judicious application of mechanical energy could address this challenge. Specifically a vortex fluidic device (VFD) directs controlled mechanical energy into solution to accelerate enzyme-catalyzed reactions [29]. This thin film microfluidic platforms can disrupt membranes and drive protein folding [30], and potentially assist with the deaggregation, and solubilization of formalin-fixed samples in addition to acceleration of enzyme activity. Here, we tested the efficacy of using a VFD to accelerate proteinase K activity, and increase the process throughput and efficiency of extracting DNA from formalin-fixed specimens (Fig 2).

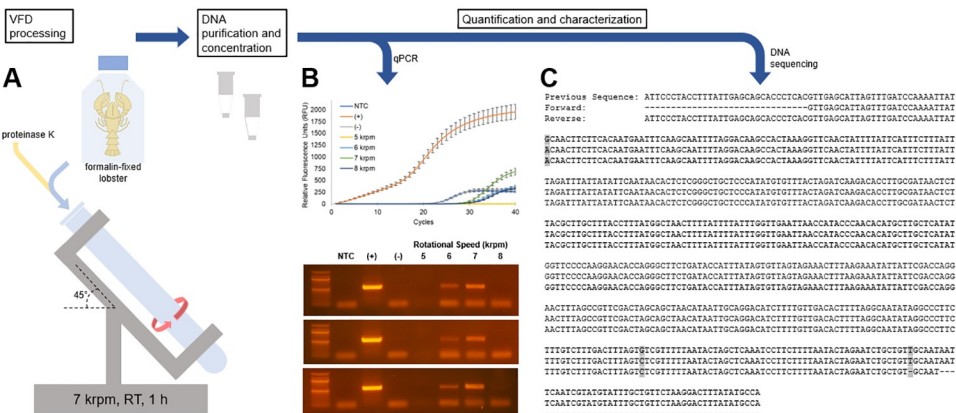

**Fig 2. Schematic of the VFD-mediated fDNA recovery technique. (A)** The protocol begins with Vortex Fluidic Device (VFD) treatment (7 krpm, room temperature, abbreviated RT, 1 h) of a mixture of proteinase K and the frozen, then broken-up tissue. The reaction mixture is next processed to remove solids and DNA polymerase inhibitors. The recovered fDNA is then purified and concentrated. Finally, the DNA is amplified, quantified, and characterized by **(B)** qPCR and **(C)** DNA sequencing of the samples. Larger versions of panels **B** and **C** are provided in S1 and S2 Figs. Threshold cycle ($C_t$) and endpoint fluorescence values are given in S1 Table.

Thus, we optimized the recovery of fDNA from formalin-fixed specimens through VFD and post-processing purification. Our results suggest that this method recovers fDNA with 40 to 85% greater yields than conventional methods without requiring harsh conditions, and can decrease treatment time from days to hours.

# Protocol

## Reagents

- Proteinase K (Promega, cat. no. V3021, lyophilized)

- Sodium dodecyl sulphate (SDS) (Fisher Bioreagents, cat no. BP8200-5)

- Tris hydrochloride (Fisher Bioreagents, cat. no. BP153-1)

- Calcium chloride (Fisher Chemical, cat no. C614-500)

- Glycerol (ACS reagent, cat. no. G7893-4L)

- Ethylenediaminetetraacetic acid (EDTA) (Acros Organics, cat. no. 147850010)

- Nitrogen (Liquid) (Airgas Healthcare, cat. no. UNI977)

- Ethanol (200 proof, Molecular Biology Grade) (Fisher Scientific, cat. no. BP2828-500)

- Clean & Concentrator Kit (Zymo Research, cat. no. D4006)

## Equipment

- Vortex Fluidic Device v.2 (VFD) (Vortex Fluidic Technologies)

- Microliter pipettes (1000 μL, 200 μL, 20 μL, and 10 μL)

- Refrigerated, tabletop centrifuge

- Vortex mixer

- Mortar & pestle

- Hemostat

- Forceps

- Razorblades

- Eppendorf tubes (1.7 mL)

## Reagent setup

**Critical Step**: Nanopure water (ddH$_2$O) is used for all buffers and solutions. Buffers are autoclaved or sterile-filtered, if containing SDS, prior to addition of enzymes and use. Enzyme-containing solutions are stored at -80 ˚C.

- **Proteolysis buffer** 20 mM Tris-HCl, 50 mM EDTA, 1% w/v SDS, pH 8.0[31]

- **Proteinase K solution** 10 mg/mL proteinase K, 20 mM Tris-HCl, 1 mM CaCl$_2$, 50% glycerol, pH 8.0

## Procedure

### Tissue sample preparation. Timing: ≈1 h.

1. After removal of a small portion (≈5 to 10 g) of the preserved biological tissue from the preservative, immerse the tissue in liquid nitrogen until frozen, and grind with a mortar and pestle for ≈1 min. The small pieces of ground tissue are aliquoted (≈1 to 1.5 g) into Eppendorf tubes (henceforth termed tubes).
   **Safety Note**: Exercise caution when utilizing a sharp edge to prevent puncturing or cutting personal protective equipment or skin.
   **Pause Point**: At this stage, the sample can be stored at -80 ˚C for later processing.

2. On an autoclaved glass surface, mince the formalin-fixed tissue sample with a flame-sterilized razorblade held by a sterilized hemostat for ≈5 min.
   **Safety Note**: Exercise caution when utilizing a sharp edge to prevent puncturing or cutting personal protective equipment or skin.
   **Critical Step**: This step increases the surface area to volume ratio of the tissue and, thus, improves the proteinase K access to the sample.

3. To remove the preservative fluid, wash the sample three times with the proteolysis buffer (1 mL). For each wash step, briefly vortex, centrifuge (15 krcf, 3 min), and decant the samples. If required, an addition centrifugation (15 krcf, 1 min) can remove any residual buffer.

### VFD treatment. Timing: ≈1 h.

4. Transfer the ground, minced tissue to the bottom of an autoclaved 20 mm VFD sample tube. Add proteolysis buffer (950 μL) and then proteinase K solution (50 μL).

5. Seal the VFD sample tube with a rubber septum, and use the VFD to spin the sample (7 krpm, 1 h, RT).
   **Critical Step**: This rotational speed is optimal for fDNA amplification and sequencing.

### Sample post-processing and purification. Timing: ≈1.5 h.

6. Following VFD processing, immediately transfer the sample, including both the processed tissue and solution, from the VFD sample tube to a new tube.

7. Immediately, centrifuge the sample (15 krcf, 5 min, RT) to remove the tissue. Transfer the supernatant to a clean tube.

8. Incubate the supernatant on wet ice for 30 minutes.
   **Critical Step**: A white precipitate (SDS) collects at the bottom of the tube. The presence of SDS negatively affects PCR yields and subsequent purification of fDNA.

9. Immediately, centrifuge the sample (15 krcf, 10 min, 4 ˚C) to remove SDS, and transfer the supernatant to a new tube without disturbing the SDS pellet.
   **Critical Step**: The supernatant must be transferred immediately following centrifugation to prevent resolubilization of SDS.
   **Pause Point**: At this stage, the sample may be frozen at -20 or -80 ˚C for later analysis.

10. Process 400 μL of the supernatant with a Zymo DNA Clean & Concentrator Kit, according to the manufacturer's instructions.

**fDNA quantification and characterization. Timing: ≈8 h.**

11. The isolated fDNA can be used for further amplification, characterization, quantification, and sequencing.

## Materials & methods

One adult male Lobster (*H. americanus*) was purchased in February 2017 from a local lobster fishery in Boston, MA. The lobster was euthanized by quickly severing the ganglia behind the eyes with a sharp knife. The body was then placed whole in a solution of 3.7% formaldehyde in 0.9 M phosphate-buffered saline (which approximates the salinity of seawater). The lobster was maintained at room temperature for one month, and then shipped to the University of California Irvine in March 2017. All lobster trails shown here have used muscle recovered from the chelipeds (primary claws), which have remained in formalin for the two-year duration of this study. For experimental treatments, the lobster claw tissue was processed according to the procedure described above.

Proteinase K (Promega, V3021, lyophilized) was solubilized and diluted to 10 mg/mL in storage buffer (20 mM Tris-HCl, 1 mM $CaCl_2$, 50% glycerol, pH 8.0).

For DNA isolation, 100 (±1) mg of lobster tissue samples were utilized. Tissue preparation, VFD processing, and DNA purification used methods described above. In the experiments reported here, the VFD was operated in the confined, not continuous flow, mode [32–33]; specifically, 1 mL volumes were used. The negative control samples consisted of identical tissue samples and treatment but were not subjected to VFD processing. A positive control contained VFD-processed fresh lobster tissue. Additionally, fixed and fresh lobster tissue were processed overnight at 56 °C without the VFD for the conventional method controls, based on Table 1. Finally, intermediate controls combined the conventional and VFD-mediated methods to minimize the number of variables changed per experiment. These intermediate controls consisted of fixed and fresh lobster processed overnight (as in the conventional method) at room temperature (as in the VFD-mediated method) without use of the VFD.

The positive controls for PCR quantification were DNA obtained from fresh (non-formalin-fixed), ground lobster claw tissue. The DNA recovery from this sample applied Chelex 100 Resin (Bio-Rad, 10% w/v in 500 μL in $ddH_2O$) with the manufacturer's protocol. The mixture of a tissue fragment and resin was vortexed and centrifuged briefly before incubation at 90 to 95 °C for 20 to 35 min. Following another brief vortexing and centrifugation, the supernatant was isolated as the positive control for lobster fDNA.

DNA extraction yields were compared by quantitative PCR (qPCR) (Bio-Rad iCycler). For PCR, reaction mixtures (10 μL) applied the Phusion DNA polymerase (0.2 U final concentration, New England Biolabs) and buffer (5× diluted final concentration, New England Biolabs), DMSO (10% v/v final concentration), dNTPs (0.5 mM each final concentration, New England Biolabs), primers (8–33 ng each final concentration, Integrated DNA Technologies) (Table 2), and SYBR Green I dye (10,000× diluted final concentration, Thermo Fisher Scientific). A PCR

**Table 2. PCR primer sequences and annealing temperatures.**

| Gene Target: mitochondrial ATP synthase | | Primer | Annealing Temperature (est.) (°C) |
|---|---|---|---|
| 579 bp | Forward | GGGTTACTTTTTATTCCCTACCTTTATTGAGC | 60 |
| | Reverse | GGCATATAAAGTCCTTAGAACAGCAAATACATACG | |
| 183 bp | Forward | GGGTTACTTTTTATTCCCTACCTTTATTGAGC | 60 |
| | Reverse | CAGCCCGAGAGTGTTATTGAATATAATAAATC | |

was performed with 1 cycle of 94 ˚C for 5 min followed by 40 to 50 cycles of 94 ˚C for 1 min, 60 ˚C for 1 min, and 72 ˚C for 2 min, followed by 1 cycle of 72 ˚C for 5 min.

The recovered fDNA from the formalin-fixed lobster was quantified by UV-Vis absorbance. The absorbance spectra of the samples, diluted in ddH$_2$O (1:50), were measured (Jasco V-730 Spectrophotometer). The spectra were recorded from 200 to 400 nm, in triplicate (technical replicates), with a scanning speed of 200 nm/min and intervals of 0.5 nm with ddH$_2$O as the blank. Two buffer only controls examined the reaction mixture without lobster tissue. The first was processed in the VFD (7.5 krpm, 1 h, RT), and the second did not use VFD processing. The 1 kb Plus DNA Ladder (10 μg, 1000 μL) (Thermo Fisher Scientific, 1.0 μg/μl) was used to estimate DNA sizes and concentrations; a positive control applied DNA from fresh lobster tissue extracted with Chelex 100 Resin, as described above.

A SYBR Green I fluorescence assay also quantified the dsDNA concentration [34]. To derive a DNA concentration calibration curve, a 1 kb Plus DNA Ladder (Thermo Fisher Scientific, 1.0 μg/μL) was diluted to concentrations of 0 ng/μL to 1.25 ng/μL. These dilutions (100 μL) and SYBR Green I dye (100 μL, Thermo Fisher Scientific, 1250× diluted in ddH$_2$O) were added to a black, clear-bottom 96-well plate (Corning, 3615). The fluorescence of each well was measured ($\lambda_{ex}$ 485 nm, $\lambda_{ex}$ 550 nm, Bandwidth 20 nm, Gain 50). The fluorescence of fDNA from the fixed lobster samples (100× diluted) were similarly measured, and the dsDNA concentrations were estimated using the calibration curve shown in S4 Fig.

## Results and discussion

Conventional methods applying proteinase K to remove protein-DNA crosslinks require >17 h [26–27], and typically the reaction runs for >1 day. Here we report enzyme acceleration techniques via VFD mechanical stimulation to enable DNA recovery in <2 h from lobster claw tissue (*H. americanus*) preserved in formalin (3.7% formaldehyde in phosphate-buffered saline, a condition iso-osmotic with seawater). Key variables requiring optimization included mechanical breakage of the tissue sample, length of fDNA sequences to be amplified, and rotational speed of the VFD. As observed for other VFD-enhanced enzymes [29], proteinase K activity can be accelerated using a VFD. The approach allows recovery of free DNA (fDNA) from formalin-fixed samples. The rotational speed of the tube in the VFD determines the level of shear, micro-mixing, and fluid dynamics experienced by the thin film of liquid [32]. The tilt angle of the VFD can be important, and 45˚ relative to the horizontal is the optimal tilt angle for a myriad of applications, including accelerating enzymatic reactions [29, 32].

Mechanical breakage of the tissue sample emerged as a key variable for efficient and robust isolation of fDNA. Smaller fragments increase the surface area to volume ratio, allowing greater efficiency of proteinase K digestion. Also, the VFD removes solids from solution by centrifugation, and, therefore, breaking the tissue sample into small fragments improves the efficiency of VFD-mediated fDNA recovery. The formalin-fixed tissue was first frozen in liquid nitrogen then ground into small particles using a clean mortar and pestle. When extracted using traditional extraction approaches, the ground material yielded modest and inconsistent quantities of fDNA (Fig 2 and S1 Table). The lack of reproducibility suggested a need for further mechanical breakdown of the tissue. Beadbeating and sonication increased the observed breakdown of tissue, but both methods failed to improve PCR product yields. These failures may be due to the fact that sonication or the resultant heat generation through cavitation could introduce additional breaks in the DNA. However, we found that a second, mincing step with a sterile, new razorblade consistently improved fDNA recovery (Fig 4).

Also, the length of the amplified DNA proved critical for consistent recovery from formalin-fixed samples. In initial experiments, we amplified a 579 bp sequence of fDNA. The amplified

DNA was the correct lobster sequence (Figs 2 and 4, S2 Fig), but this long fragment proved difficult to amplify repeatedly. Thus, primers targeting a 183 bp sequence were used for all further experiments reported here. The shorter target amplicon decreased the chance of DNA fragmentation damaging the sequence and preventing PCR amplification. As expected, DNA sequencing quality in the non-VFD control reactions was difficult to monitor; only three out of thirteen cumulative control amplification reactions yielded fDNA detectable by gel electrophoresis.

The final parameter optimized for fDNA extraction was the rotational speed of the VFD. Previous studies with the VFD have demonstrated that acceleration of enzymatic catalysis occurs at rotational speeds between 5 and 9 krpm with the VFD at a 45° tilt angle [29]. Hypotheses attribute the enzymatic acceleration phenomena to two interconnected actions. First, the periodic change in the thickness of the thin fluid film present in the VFD, results in intense micro-mixing and high mass transfer. Second, the Faraday waves arising from this periodic change contribute to zones of high and low pressure within the reaction mixture and, accordingly, the enzyme present in solution. The pressure oscillations could increase substrate accessibility and removal of product from the enzyme active site, which also benefit from the high mass transfer of the VFD.

To identify optimal speeds for fDNA recovery, a systematic assessment of rotational speeds between 5 and 9 krpm at intervals of 1 krpm was conducted. Heterogeneity inherent to pulverized tissue imparts idiosyncratic and uncontrollable variables into this optimization and subsequent isolation of fDNA. Thus, the yields of fDNA, as determined by UV-Vis spectrophotometry, were sometimes inconsistent (It is well known that absorption of 260 nm is a proxy for DNA concentrations in solution, whereas the ratio of absorbances at 260:280 nm serves as an indicator of DNA purity). Though absorbance at 260 nm increased for the VFD-processed samples at various rotational speeds relative to the non-VFD-processed negative control, the ratio of 260:280 nm absorbances was significantly <1.8 for all rotational speeds (Fig 3 and S3 Fig), which likely indicates inclusion of protein in the VFD-treated samples [35]. The DNA-associated, absorbance at 260 nm was greatest for the samples processed at 8 krpm. This rotational speed is within the VFD-based rate enhancement zone for reactions in aqueous solvents. That said, the highest, most consistently observed PCR yields were obtained for the rotational speed of the VFD of 7 krpm (which is discussed in more detail below; see Figs 2 and 4, S1 and S5 Figs, S2 Table).

To more robustly quantify the concentration of double-stranded DNA (dsDNA) in the fDNA, a fluorescence-based DNA intercalating assay was performed using SYBR Green I dye. Samples were diluted to reach the dye's linear range, and the sample fluorescence ($\lambda_{ex}$ 485 nm, $\lambda_{em}$ 550 nm) was measured against a standard curve to determine the concentrations of the rescued dsDNA. The assay demonstrated that VFD-processed samples yielded 40 to 85% more dsDNA than the control, non-VFD processed sample (Fig 3). Thus, the VFD improved the rescue of dsDNA from formalin-fixed tissue compared to the non-VFD-treated negative control.

Moreover, and most importantly, DNA recovered via VFD-enhanced extraction were amenable to amplification via PCR and quantitative PCR (or qPCR). Post-VFD treatment, samples were purified and concentrated with a Zymo™ DNA Clean & Concentrator Kit, which removes DNA polymerase inhibitors and proteins [16, 36]. Using the fDNA samples and qPCR, we observed reproducible amplification of an 183 bp target amplicon from the gene encoding ATP synthase from samples processed with a VFD rotating at either 6 or 7 krpm (Fig 4, Table 2). Paradoxically, these speeds had lower fDNA yields when compared to the 8 krpm (as described above; see Fig 3a and 3b). However, the 8 krpm treated DNA would amplify <50% of the time (n = 8, shown in S5 Fig). Though yields were greater, the more aggressive treatment is apparently liberating other compounds that can be problematic to PCR amplification (this

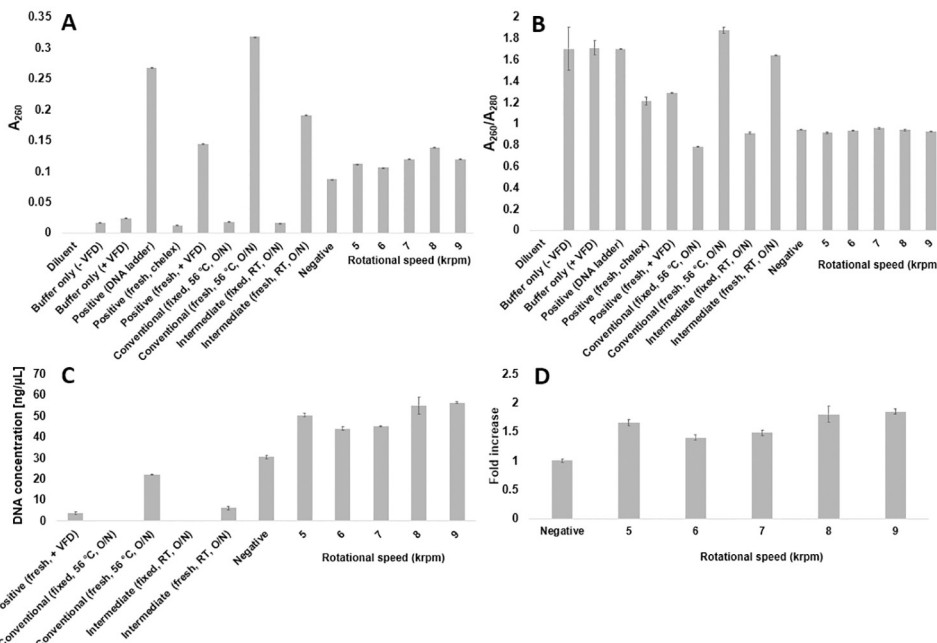

**Fig 3. Quantification for optimizing the VFD rotational speed for fDNA yields.** After proteinase K and VFD treatment at the indicated speeds, (**A**) absorbance at 260 nm and (**B**) the ratio of absorbances 260:280 nm quantifies DNA and protein yields, respectively, with positive and negative controls. Full UV-vis spectra for these samples are shown in S3 Fig. SYBR Green I fluorescence-quantified (**C**) dsDNA concentration and (**D**) fold increase in dsDNA yield between non-VFD-processed and VFD-processed samples. The negative control indicates samples not subjected to VFD processing. Buffer only controls lacked lobster tissue. The positive controls included DNA that had not been formalin fixed. Additional controls demonstrated a conventional method and an intermediate method (processing time of the conventional method and temperature of the VFD-mediated method) used to process fixed and fresh tissue without the VFD. The error bars designate the standard deviation for sample measurements at the indicated condition (technical replicates, n = 3).

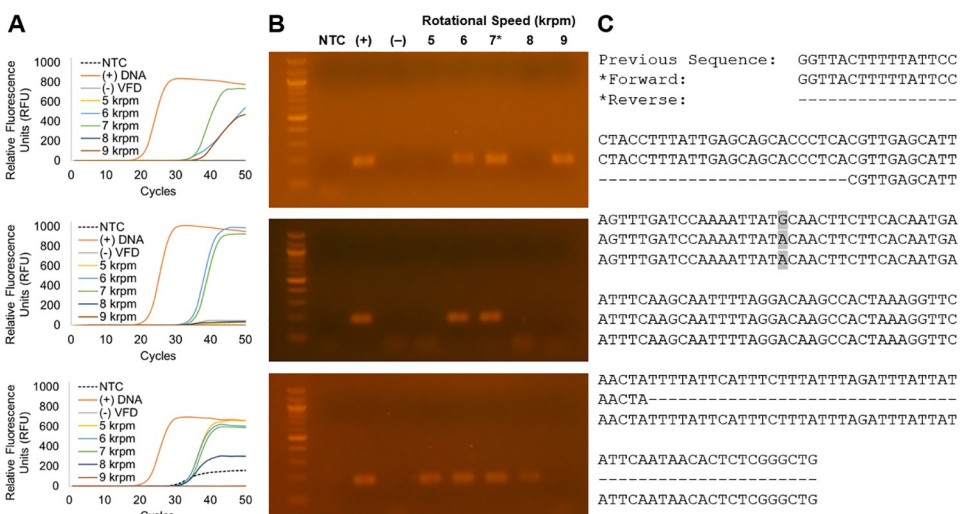

**Fig 4. Amplification of an 183-bp fDNA target from the ATP synthase gene of the lobster mitochondrial genome.** (**A**) Quantitative PCR and (**B**) agarose DNA gel electrophoresis identified 7 krpm as the optimal VFD rotational speed for qPCR amplification. Threshold cycle and endpoint fluorescence values are provided in S2 Table. The variable-rotational speed PCR reactions were compared to a no template control (NTC), a fresh lobster DNA positive control (+), and a non-VFD-processed negative control (–). (**C**) The 7 krpm VFD-processed qPCR product (*) was subjected to Sanger sequencing; a mutation (G2728A, GenBank No. HQ402925) was observed (highlighted).

phenomena has also been seen in algal cells and more) [37]. Additionally, it is plausible that greater VFD rotational speeds could result in both greater yields and further fragmentation of fDNA, which could explain the higher DNA concentrations, but lower amplification efficacy [16].

From the formalin-fixed samples, the 7 krpm VFD-processed DNA sample yielded the highest levels of fDNA amplification as measured by qPCR. For example, on average 94±4% of the endpoint positive control fluorescence signal was obtained with a low threshold cycle ($C_t$) value, averaging 35.4±0.7 cycles (S2 Table). Comparatively, the no template controls (NTC) did not surpass the threshold in two of three trials, and their fluorescence averaged 10±10% of the endpoint positive control fluorescence signal (S2 Table). The lower $C_t$ values demonstrated a greater yield of DNA. Furthermore, the PCR product of the fDNA rescued from the 7 krpm VFD-processing condition could readily be sequenced via Sanger sequencing. This sequence exhibited 99.5% homology to the expected sequence [38]. The 6 krpm-processed sample offered a decreased yield of amplified DNA. Notably, PCR amplification failed for the non-VFD-treated sample.

The data herein illustrate the efficacy of our new VFD-enabled method for fDNA recovery. We have demonstrated the successful amplification of fDNA from biological specimens treated with formaldehyde. We have also shown that fDNA can be used to great effect with appropriately designed qPCR assays. Therefore, we are optimistic that this method presents a potentially valuable method for increasing the throughput of fDNA recovery. Increasing the rate at which fDNA can be recovered is a timely pursuit, as there are tens of millions of formalin-fixed samples stored in museums around the world. These organisms provide a "time capsule" of sorts, revealing the genomic adaptations of organisms to a pre-industrial world. Indeed, museum specimens may become our first tool for understanding the extent to which anthropogenic factors are shaping our biosphere. It is also important to note that fDNA is, unfortunately, subject to irreparable damage and fragmentation, and any PCR-based amplification from such samples will always be highly dependent on rescue conditions, primer design (especially amplicon length), tissue mincing, and other factors. The VFD method here may help investigators tap into this enormous genetic repository.

## Supporting information

**S1 Fig. Expanded versions of Fig 2B. (A)** qPCR was used to quantify DNA encoding ATP synthase (579 bp) recovered from formalin-fixed, frozen and ground lobster tissue by VFD-processing (1 h, RT) and non-VFD (–) methods (1 h, 37 ˚C) at the indicated rotational speeds. DNA isolated from fresh lobster tissue provided the positive control. **(B)** Following qPCR, the resultant DNA was visualized by 1% agarose gel electrophoresis, which highlights the formation of primer dimers in the negative controls, including for the negative control with omitted template (no template control, NTC). Error bars indicate standard deviation (technical replicate, n = 3).
(PDF)

**S2 Fig. Expanded version of Fig 2C.** Sanger DNA sequencing of the target mitochondrial ATP synthase sequence from ground and sonicated then VFD-processed (9 krpm, 1 h, RT), formalin-fixed lobster tissue. Within the range of Sanger sequencing accuracy, two mutations (G2728A & G3136C, GenBank No. HQ402925) were observed as indicated. The reference sequence has been described previously [1].
(PDF)

**S3 Fig. The absorbance spectra of reaction supernatants for the fixed lobster samples and positive and negative controls given in Fig 3.**
(PDF)

**S4 Fig. The calibration curve for DNA quantification by the SYBR Green I intercalation fluorescence assay.**
(PDF)

**S5 Fig. qPCR and PCR (***) of fDNA with (A) 183 bp ATP synthase amplicon primers showing multiple experiments to illustrate experimental consistency for the optimal VFD conditions reported here, but not 8 krpm rotational speeds, (B) 579 bp ATP synthase amplicon primers, and (C) 549 bp NADH dehydrogenase amplicon primers (forward: TCATCCATAGCACCAACCTTC; reverse: TGTTCAAGGCACTCTTATTTATATG; annealing temperature: 61 ˚C).**
(PDF)

**S1 Table. Threshold cycle values (Ct) and endpoint fluorescence values of qPCR with the fDNA (Fig 2, S1 and S2 Figs)***.
(PDF)

**S2 Table. Threshold cycle values (Ct) and endpoint fluorescence values of qPCR with the fDNA (Fig 4)***.
(PDF)

## Acknowledgments

We gratefully thank UCI's Department of Molecular Biology and Biochemistry for access to the qPCR thermocycler.

## Author Contributions

**Conceptualization:** Peter R. Girguis, Gregory A. Weiss.

**Data curation:** Christian A. Totoiu.

**Formal analysis:** Christian A. Totoiu, Peter R. Girguis, Colin L. Raston, Gregory A. Weiss.

**Funding acquisition:** Christian A. Totoiu, Peter R. Girguis, Colin L. Raston, Gregory A. Weiss.

**Investigation:** Christian A. Totoiu, Aspen T. Reese, Gregory A. Weiss.

**Methodology:** Christian A. Totoiu, Aspen T. Reese, Sudipta Majumdar, Colin L. Raston, Gregory A. Weiss.

**Project administration:** Colin L. Raston, Gregory A. Weiss.

**Supervision:** Sudipta Majumdar, Peter R. Girguis, Colin L. Raston, Gregory A. Weiss.

**Validation:** Jessica M. Phillips, Sudipta Majumdar.

**Visualization:** Christian A. Totoiu, Gregory A. Weiss.

**Writing – original draft:** Christian A. Totoiu, Gregory A. Weiss.

**Writing – review & editing:** Christian A. Totoiu, Jessica M. Phillips, Aspen T. Reese, Sudipta Majumdar, Peter R. Girguis, Colin L. Raston, Gregory A. Weiss.

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
