## [Decision Letter · Decision Letter 0]

8 Jan 2020

Vortex fluidics-mediated DNA rescue from formalin-fixed museum specimens

PONE-D-19-31335

Dear Dr. Weiss,

We are pleased to inform you that your manuscript has been judged scientifically suitable for publication and will be formally accepted for publication once it complies with all outstanding technical requirements.

With kind regards,

Andrew Baggaley, Ph.D.

Academic Editor

PLOS ONE

Journal Requirements:

1. We note that you have included the phrase “data not shown” in your manuscript. Unfortunately, this does not meet our data sharing requirements. PLOS does not permit references to inaccessible data. We require that authors provide all relevant data within the paper, Supporting Information files, or in an acceptable, public repository. Please add a citation to support this phrase or upload the data that corresponds with these findings to a stable repository (such as Figshare or Dryad) and provide and URLs, DOIs, or accession numbers that may be used to access these data. Or, if the data are not a core part of the research being presented in your study, we ask that you remove the phrase that refers to these data.

2. We note that you have a patent relating to material pertinent to this article. Please provide an amended statement of Competing Interests to declare this patent (with details including name and number), along with any other relevant declarations relating to employment, consultancy, patents, products in development or modified products etc.

Please confirm that this does not alter your adherence to all PLOS ONE policies on sharing data and materials, as detailed online in our guide for authors http://journals.plos.org/plosone/s/competing-interests by including the following statement: "This does not alter our adherence to  PLOS ONE policies on sharing data and materials.” If there are restrictions on sharing of data and/or materials, please state these. Please note that we cannot proceed with consideration of your article until this information has been declared.

Please respond by return email and we will update the online submission form on your behalf.

Reviewers' comments:

Reviewer's Responses to Questions

**Comments to the Author**

1. Is the manuscript technically sound, and do the data support the conclusions?

Reviewer #1: Yes

2. Has the statistical analysis been performed appropriately and rigorously? 

Reviewer #1: Yes

3. Have the authors made all data underlying the findings in their manuscript fully available?

Reviewer #1: Yes

4. Is the manuscript presented in an intelligible fashion and written in standard English?

Reviewer #1: Yes

5. Review Comments to the Author

Reviewer #1: The manuscript is clear and well written, and the results easily repeatable from the detail provided. The authors address an important issue that is of increasing interest and hopefully becoming technologically more and more feasible as methods develop. If we can find methodology that would allow regular and at least semi-consistent amplification of DNA from formalin fixed tissues, it would absolutely change the way we can use the extensive museum collections available around the world and the questions we can ask about biology of the animals preserved there. Because of this, the paper is significant in that it contributes substantially to the testing of methods to optimize recovery of DNA from formalin fixed specimens. The methods would require substantial refinement to work on most formalin fixed material given the large amount of tissue tested 1 g of muscle and the short duration of time the tested tissue was in formalin, but none the less the contribution and idea are significant and should be published. They have provided an important first step toward refining methods for using formalin fixed specimens for genetic studies and should be published without any major revisions. It was not clear what the contributions of the many authors were and why such a simple and straightforward project required so many authors, but that is up to them to decide who contributed to the study.

6. PLOS authors have the option to publish the peer review history of their article (what does this mean?). If published, this will include your full peer review and any attached files.

Reviewer #1: No

---

## [Editor Report · Acceptance letter]

14 Jan 2020

PONE-D-19-31335 

Vortex fluidics-mediated DNA rescue from formalin-fixed museum specimens 

Dear Dr. Weiss:

I am pleased to inform you that your manuscript has been deemed suitable for publication in PLOS ONE. Congratulations! Your manuscript is now with our production department. 

With kind regards,

on behalf of

Dr Andrew Baggaley 

Academic Editor

PLOS ONE